# Convolutional Monge Mapping Normalization for learning on sleep data

**Theo Gnassounou**
Université Paris-Saclay, Inria, CEA
Palaiseau 91120, France
`theo.gnassounou@inria.fr`

**Rémi Flamary**
IP Paris, CMAP, UMR 7641
Palaiseau 91120, France
`remi.flamary@polytechnique.edu`

**Alexandre Gramfort**[*]
Université Paris-Saclay, Inria, CEA
Palaiseau 91120, France
`alexandre.gramfort@inria.fr`

## Abstract

In many machine learning applications on signals and biomedical data, especially electroencephalogram (EEG), one major challenge is the variability of the data across subjects, sessions, and hardware devices. In this work, we propose a new method called Convolutional Monge Mapping Normalization (CMMN), which consists in filtering the signals in order to adapt their power spectrum density (PSD) to a Wasserstein barycenter estimated on training data. CMMN relies on novel closed-form solutions for optimal transport mappings and barycenters and provides individual test time adaptation to new data without needing to retrain a prediction model. Numerical experiments on sleep EEG data show that CMMN leads to significant and consistent performance gains independent from the neural network architecture when adapting between subjects, sessions, and even datasets collected with different hardware. Notably our performance gain is on par with much more numerically intensive Domain Adaptation (DA) methods and can be used in conjunction with those for even better performances.

## 1 Introduction

**Data shift in biological signals** Biological signals, such as electroencephalograms (EEG), often exhibit a significant degree of variability. This variability arises from various factors, including the recording setup (hardware specifications, number of electrodes), individual human subjects (variations in anatomies and brain activities), and the recording session itself (electrode impedance, positioning). In this paper, we focus on the problem of sleep staging which consists of measuring the activities of the brain and body during a session (here one session is done over a night of sleep) with electroencephalograms (EEG), electrooculograms (EOG), and electromyograms (EMG) [1] to classify the sleep stages. Depending on the dataset, the populations studied may vary from healthy cohorts to cohorts suffering from disease [2, 3, 4]. Different electrodes positioned in the front/back [3] or the side of the head [4] can be employed. Sleep staging is a perfect problem for studying the need to adapt to this variability that is also commonly denoted as data shift between domains (that can be datasets, subjects, or even sessions).

**Normalization for data shift** A traditional approach in machine learning to address data shifts between domains is to apply data normalization. Different approaches exist in the literature to

---

[*]A. Gramfort joined Meta and can be reached at `agramfort@meta.com`

normalize data. One can normalize the data per `Session` which allows to keep more within-session variability in the data [5]. If the variability during the session is too large, one can normalize independently each window of data (*e.g.,* 30 s on sleep EEG) [6], denoted as `Sample` in the following. It is also possible not to normalize the data, and let a neural network learn to discard non-pertinent variabilities helped with batch normalization [7, 8]. More recently, a number of works have explored the possibility of learning a normalization layer that is domain specific in order to better adapt their specificities [9, 10, 11, 12, 13, 14, 15]. However, this line of work usually requires to have labeled data from all domains (subjects) for learning which might not be the case in practice when the objective is to automatically label a new domain without an expert.

**Domain Adaptation (DA) for data shift**   Domain Adaptation is a field in machine learning that aims at adapting a predictor in the presence of data shift but in the absence of labeled data in the target domain. The goal of DA is to find an estimator using the labeled source domains which generalizes well on the shifted target domain [16]. In the context of biological signals, DA is especially relevant as it has the ability to fine-tune the predictor for each new target domain by leveraging unlabeled data. Modern DA methods, inspired by successes in computer vision, usually try to reduce the shift between the embeddings of domains learned by the feature extractor. To do that, a majority of the methods try to minimize the divergence between the features of the source and the target data. Several divergences can be considered for this task such as correlation distance [17], adversarial method [18], Maximum Mean Discrepancy (MMD) distance [19] or optimal transport [20, 21]. Another adaptation strategy consists in learning domain-specific batch normalization [9] in the embedding. Interestingly those methods share an objective with the normalization methods: they aim to reduce the shift between datasets. To this end, DA learns a complex invariant feature representation whereas normalization remains in the original data space. Finally, test-time DA aims at adapting a predictor to the target data without access to the source data [22], which might not be available in practice due to privacy concerns or the memory limit of devices.

**Contributions**   In this work we propose a novel and efficient normalization approach that can compensate at test-time for the spectral variability of the domain signals. Our approach called Convolutional Monge Mapping Normalization (`CMMN`), illustrated in Figure 1, uses a novel closed-form to estimate a meaningful barycenter from the source domains. Then `CMMN` uses a closed-form solution for the optimal transport Monge mapping between Gaussian random signals to align the power spectrum density of each domain (source and target) to the barycenter. We emphasize that `CMMN` is, to the best of our knowledge, the first approach that can adapt to complex spectral shifts in the data without the need to access target datasets at training time or train a new estimator for each new domain (which are the limits of DA).

We first introduce in section 2 the problem of optimal transport (OT) between Gaussian distributions, and then propose a novel closed-form solution for the Wasserstein barycenter for stationary Gaussian random signals. We then use this result to propose our convolutional normalization procedure in section 3, where implementation details and related works are also discussed. Finally section 4 reports a number of numerical experiments on sleep EEG data, demonstrating the interest of `CMMN` for adapting to new subjects, sessions, and even datasets, but also study its interaction with DA methods.

**Notations**   Vectors are denoted by small cap boldface letters (*e.g.,* $\mathbf{x}$), and matrices are denoted by large cap boldface letters (*e.g.,* $\mathbf{X}$). The element-wise product is denoted by $\odot$. The element-wise power of $n$ is denoted by $\cdot^{\odot n}$. $[K]$ denotes the set $\{1, ..., K\}$. $|.|$ denotes the absolute value. The discrete convolution operator between two signals is denoted as $*$. Any parameters written with a small $k$ (*e.g.,* $\mathbf{X}_k$ or $\mathbf{x}_i^k$) is related to the source domain $k$ with $1 \leq k \leq K$. Any parameters written with a small t (*e.g.,* $\mathbf{X}_t$ or $\mathbf{x}_i^t$) is related to the target domain.

## 2   Signal adaptation with Optimal Transport

In this section, we first provide a short introduction to optimal transport between Gaussian Distributions, and then discuss how those solutions can be computed efficiently on stationary Gaussian signals, exhibiting a new closed-form solution for Wasserstein barycenters.

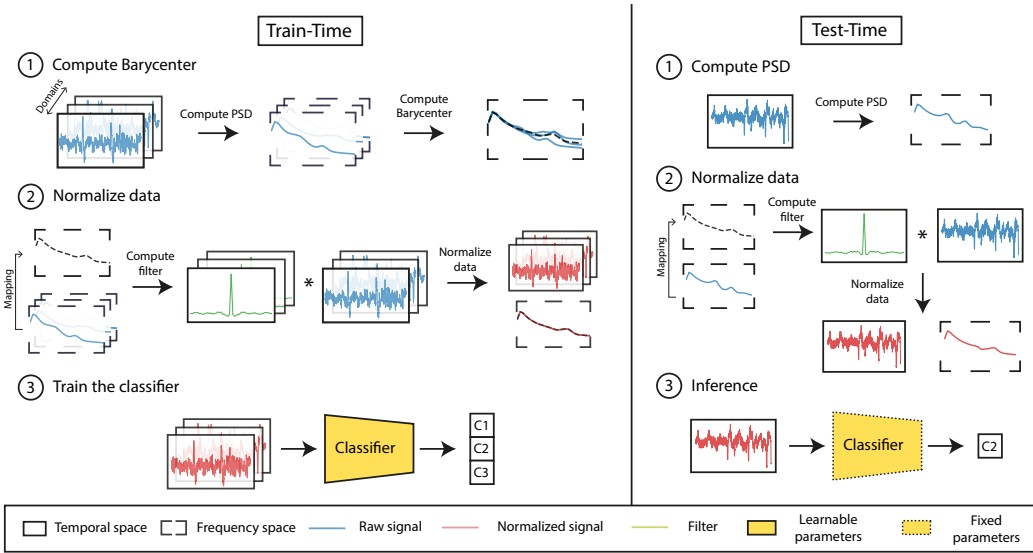

Figure 1: Illustration of the `CMMN` approach. At train-time the Wasserstein barycenter is estimated from 3 subjects/domains. The model is learned on normalized data. At test time the same barycenter is used to normalize test data and predict.

## 2.1 Optimal Transport between Gaussian distributions

**Monge mapping for Gaussian distributions** Let two Gaussian distributions $\mu_s = \mathcal{N}(\mathbf{m}_s, \boldsymbol{\Sigma}_s)$ and $\mu_t = \mathcal{N}(\mathbf{m}_t, \boldsymbol{\Sigma}_t)$, where $\boldsymbol{\Sigma}_s$ and $\boldsymbol{\Sigma}_t$ are symmetric positive definite covariances matrices. OT between Gaussian distributions is one of the rare cases where there exists a closed-form solution. The OT cost, also called the Bures-Wasserstein distance when using a quadratic ground metric, is [23, 24]

$$\mathcal{W}_2^2(\mu_s, \mu_t) = \|\mathbf{m}_s - \mathbf{m}_t\|_2^2 + \mathrm{Tr}\left(\boldsymbol{\Sigma}_s + \boldsymbol{\Sigma}_t - 2\left(\boldsymbol{\Sigma}_t^{\frac{1}{2}}\boldsymbol{\Sigma}_s\boldsymbol{\Sigma}_t^{\frac{1}{2}}\right)^{\frac{1}{2}}\right), \tag{1}$$

where the second term is called the Bures metric [25] between positive definite matrices. The OT mapping, also called Monge mapping, can be expressed as the following affine function :

$$m(\mathbf{x}) = \mathbf{A}(\mathbf{x} - \mathbf{m}_s) + \mathbf{m}_t, \quad \text{with} \quad \mathbf{A} = \boldsymbol{\Sigma}_s^{-\frac{1}{2}}\left(\boldsymbol{\Sigma}_s^{\frac{1}{2}}\boldsymbol{\Sigma}_t\boldsymbol{\Sigma}_s^{\frac{1}{2}}\right)^{\frac{1}{2}}\boldsymbol{\Sigma}_s^{-\frac{1}{2}} = \mathbf{A}^{\mathsf{T}}. \tag{2}$$

In practical applications, one can estimate empirically the means and covariances of the two distributions and plug them into the equations above. Interestingly, in this case, the concentration of the estimators has been shown to be in $O(N^{-1/2})$, where $N$ is the number of samples, for the divergence estimation [26] and for the mapping estimation [27]. This is particularly interesting because optimal transport in the general case is known to be very sensitive to the curse of dimensionality with usual concentrations in $O(N^{-1/D})$ where $D$ is the dimensionality of the data [28].

**Wasserstein barycenter between Gaussian distributions** The Wasserstein barycenter that searches for an average distribution can also be estimated between multiple Gaussian distributions $\mu_k$. This barycenter $\bar{\mu}$ is expressed as

$$\bar{\mu} = \arg\min_{\mu} \frac{1}{K}\sum_{k=1}^{K}\mathcal{W}_2^2(\mu, \mu_k). \tag{3}$$

Interestingly, the barycenter is still a Gaussian distribution $\bar{\mu} = \mathcal{N}(\bar{\mathbf{m}}, \bar{\boldsymbol{\Sigma}})$ [29]. Its mean $\bar{\mathbf{m}} = \frac{1}{K}\sum_k \mathbf{m}_k$ can be computed as an average of the means of the Gaussians, yet there is no closed-form for computing the covariance $\bar{\boldsymbol{\Sigma}}$. In practical applications, practitioners often use the following optimality condition from [29]

$$\bar{\boldsymbol{\Sigma}} = \frac{1}{K}\sum_{k=1}^{K}\left(\bar{\boldsymbol{\Sigma}}^{\frac{1}{2}}\boldsymbol{\Sigma}_k\bar{\boldsymbol{\Sigma}}^{\frac{1}{2}}\right)^{\frac{1}{2}}, \tag{4}$$

in a fixed point algorithm that consists in updating the covariance [30] using equation (4) above until convergence. Similarly to the distance estimation and mapping estimation, statistical estimation of the barycenter from sampled distribution has been shown to have a concentration in $O(N^{-1/2})$ [31].

## 2.2 Optimal Transport between Gaussian stationary signals

We now discuss the special case of OT between stationary Gaussian random signals. In this case, the covariance matrices are Toeplitz matrices. A classical assumption in signal processing is that for a long enough signal, one can assume that the signal is periodic, and therefore the covariance matrix is a Toeplitz circulant matrix. The circulant matrix can be diagonalized by the Discrete Fourier Transform (DFT) $\mathbf{\Sigma} = \mathbf{F}\mathrm{diag}(\mathbf{p})\mathbf{F}^*$, with $\mathbf{F}$ and $\mathbf{F}^*$ the Fourier transform operator and its inverse, and $\mathbf{p}$ the Power Spectral Density (PSD) of the signal.

**Welch PSD estimation**    As discussed above, there is a direct relation between the correlation matrix of a signal and its PSD. In practice, one has access to a matrix $\mathbf{X} \in \mathbb{R}^{N \times T}$ containing $N$ signals of length $T$ that are samples of Gaussian random signals. In this case, the PSD $\mathbf{p}$ of one random signal can be estimated using the Welch periodogram method [32] with $\hat{\mathbf{p}} = \frac{1}{N} \sum_{i=1}^{N} |\mathbf{F}\mathbf{x}_i|^{\odot 2}$ where $|\cdot|$ is the element-wise magnitude of the complex vector.

**Monge mapping between two Gaussian signals**    The optimal transport mapping between Gaussian random signals can be computed from (2) and simplified by using the Fourier-based eigen-factorization of the covariance matrices. The mapping between two stationary Gaussian signals of PSD respectively $\mathbf{p}_s$ and $\mathbf{p}_t$ can be expressed with the following convolution [27]:

$$m(\mathbf{x}) = \mathbf{h} * \mathbf{x} \,, \quad \text{with} \quad \mathbf{h} = \mathbf{F}^* \left( \mathbf{p}_t^{\odot \frac{1}{2}} \odot \mathbf{p}_s^{\odot -\frac{1}{2}} \right) \,. \tag{5}$$

Note that the bias terms $\mathbf{m}_s$ and $\mathbf{m}_t$ do not appear above because one can suppose in practice that the signals are centered (or have been high-pass filtered to be centered), which means that the processes are zero-mean. The Monge mapping is a convolution with a filter $\mathbf{h}$ that can be efficiently estimated from the PSD of the two signals. It was suggested in [27] as a Domain Adaptation method to compensate for convolutional shifts between datasets. For instance, it enables compensation for variations such as differing impedances, which can be physically modeled as convolutions [33]. Nevertheless, this paper focused on theoretical results, and no evaluation on real signals is reported. Moreover, the method proposed in [27] cannot be used between multiple domains (as explored here). This is why in the following we propose a novel closed-form for estimating a barycenter of Gaussian signals that we will use for the normalization of our CMMN method.

**Wasserstein barycenter between Gaussian signals**    As discussed above, there is no known closed-form solution for a Wasserstein barycenter between Gaussian distributions. Nevertheless, in the case of stationary Gaussian signals, one can exploit the structure of the covariances to derive a closed-form solution that we propose below.

**Lemma 1** *Consider $K$ centered stationary Gaussian signals of PSD $\mathbf{p}_k$ with $k \in [K]$, the Wasserstein barycenter of the $K$ signals is a centered stationary Gaussian signal of PSD $\bar{\mathbf{p}}$ with:*

$$\bar{\mathbf{p}} = \left( \frac{1}{K} \sum_{k=1}^{K} \mathbf{p}_k^{\odot \frac{1}{2}} \right)^{\odot 2} \,. \tag{6}$$

PROOF.    The proof is a direct application of the optimality condition (4) of the barycenter. The factorized covariances in (4), the matrix square root and the inverse can be simplified as element-wise square root and inverse, recovering equation (11). We provide a detailed proof in the appendix.

The closed-form solution is notable for several reasons. First, it is a novel closed-form solution that avoids the need for iterative fixed-point algorithms and costly computations of matrix square roots. Secondly, the utilization of the Wasserstein space introduces an alternative approach to the conventional $\ell_2$ averaging of Power Spectral Density (PSD). This approach involves employing the square root, like the Hellinger distance [34], potentially enhancing robustness to outliers. Note that while other estimators for PSD averaging could be used this choice is motivated here by the fact that we use OT mappings and that the barycenter above is optimal *w.r.t.* those OT mappings.

# 3 Multi-source DA with Convolutional Monge Mapping Normalization

We now introduce the core contribution of the paper, that is an efficient method that allows to adapt to the specificities of multiple domains and train a predictor that can generalize to new domains at test time without the need to train a new model. We recall here that we have access to $K$ labeled source domains $(\mathbf{X}_k, \mathbf{y}_k)_k$. We assume that each domain contains $N_k$ centered signals $\mathbf{x}_i^k$ of size $T$.

CMMN **at train time**   The proposed approach, illustrated in Figure 1 and detailed in Algorithm 1, consists of the following steps:

1. Compute the PSD $\hat{\mathbf{p}}_k$ for each source domain and use them to estimate the barycenter $\hat{\mathbf{p}}$ with (11).
2. Compute the convolutional mapping $\mathbf{h}_k$ (5) between each source domain and the barycenter $\bar{\mathbf{p}}$.
3. Train a predictor on the normalized source data using the mappings $\mathbf{h}_k$:

$$\min_f \quad \sum_{k=1}^{K} \sum_{i=1}^{N_k} L\left(y_i^k, f(\mathbf{h}_k * \mathbf{x}_i^k)\right) \ . \tag{7}$$

In order to keep notations simple, we consider here the case for a single sensor, but CMMN can be extended to multi-sensor data by computing independently the Monge mapping for each sensor. Note that steps 1 and 2 can be seen as a pre-processing and are independent of the training of the final predictor, so CMMN can be used as pre-processing on any already existing learning framework.

CMMN **at test time**   At test time, one has access to a new unlabeled target domain $(\mathbf{X}_t)$ and the procedure, detailed in Algorithm 2, is very simple. One can estimate the PSD $\hat{\mathbf{p}}_t$ from the target domain unlabeled data and compute the mapping $\mathbf{h}_t$ to the barycenter $\bar{\mathbf{p}}$. Then the final predictor for the new domain is $\hat{f}^t(\mathbf{x}_t) = f(\mathbf{h}_t * \mathbf{x}_t)$, that is the composition of the domain-specific mapping to the barycenter of the training data, and the already trained predictor $f$. This is a very efficient test-time adaptation approach that only requires an estimation of the target domain PSD that can be done with few unlabeled target samples. Yet, it allows for a final predictor to adapt to the spectral specificities of new domains thanks to the convolutional Monge normalization.

**Numerical complexity and filter size**   The numerical complexity of the method is very low as it only requires to compute the PSD of the domains and the barycenter in $O\left(\sum_k^K N_k T \log(T)\right)$. It is also im-

---

**Algorithm 1:** Train-Time CMMN

**Input:** $f$, $F$, $\{\mathbf{X}_k\}_k^K$
**for** $k = 1 \to K$ **do**
  $\mid$  $\hat{\mathbf{p}}_k \leftarrow$ Welch PSD estimation of $\mathbf{X}_k$
**end**
$\bar{\mathbf{p}} \leftarrow$ Compute barycenter with (11)
**for** $k = 1 \to K$ **do**
  $\mid$  $\mathbf{h}_k \leftarrow$ Compute mapping from (5)
**end**
$\hat{f} \leftarrow$ Train on adapted data with (7)
**return** $\hat{f}, \bar{\mathbf{p}}$

---

**Algorithm 2:** Test-Time CMMN

**Input:** $\hat{f}$, $\bar{\mathbf{p}}$, $\mathbf{X}_t$
$\hat{\mathbf{p}}_t \leftarrow$ Welch PSD estimation of $\mathbf{X}_t$
$\mathbf{h}_t \leftarrow$ compute mapping from 5
**return** $\hat{\mathbf{y}}_t = \hat{f}(\mathbf{h}_k^t * \mathbf{X}_k^t)$

---

portant to note that in practice, the method allows for a size of normalization filter $F$ that is different (smaller) than the size $T$ of the signals. This consists in practice in estimating the PSD using Welch periodogram on signal windows of size $F \leq T$ that can be extracted from the raw signal or from already extracted fixed sized source training samples. Indeed, if we use $F = T$ then the estimated average PSD can be perfectly adapted by the mapping, yet using many parameters can lead to overfitting which can be limited using a smaller filter size $F \leq T$, In fact, it is interesting to note that the special case $F = 1$ boils down to a scaling of the whole signal similar to what is done with a simple z-score operation. This means that the filter size $F$ is an hyperparameter that can be tuned on the data. From an implementation point of view, one can use the Fast Fourier Transform (FFT) to compute the convolution (for large filters) or the direct convolution, which both have very efficient implementation on modern hardware (CPU and GPU).

**Related Works**   CMMN is a computationally efficient approach that benefits from a wide array of recent results in optimal transport and domain adaptation. The idea of using optimal transport to adapt distributions was first proposed in [35]. The idea to compute a Wasserstein barycenter of distributions

from multiple domains and use it for adaptation was introduced in [21]. Both of those approaches have shown encouraging performances but were strongly limited by the numerical complexity of solving the OT problems (mapping and barycenters) on large datasets ($O(\sum_{k=1}^{K} N_k^3 \log(N_k))$) or at least quadratic in $N_k$ for entropic OT). CMMN does not suffer from this limitation as it relies on both Gaussian and stationary signals assumptions that allow to estimate all the parameters for a complexity $O(\sum_{k=1}^{K} N_k \log(N_k))$ linear with the number of samples $N_k$, and quasi-linear in dimensionality of the signals $T$. The use of Gaussian modeling and convolutional Monge mapping for DA was first proposed in [27] but the paper was mostly theoretical and only focused on the standard 2-domain DA problem whereas CMMN handles multi-source and provides test-time adaptation.

Finally, CMMN also bears resemblance with the convolutional normalization layer proposed in [11] that also uses the FFT for fast implementation. Yet, it needs to be trained using labeled source and target data, which prevents its use on DA at test time on new unseen domains.

## 4 Numerical experiments

In this section, we evaluate CMMN on the clinical application of sleep stage classification from EEG signals with [6, 36]. In the following one domain can be a session of one subject (*i.e.,* in the domain-specific experiment in section 4.2) or one subject (*i.e.,* all other experiments). We first compare CMMN to classical normalization methods and to subject-specific normalizations. Next, we illustrate the behavior of CMMN when used with different neural network architectures, and study the effect of CMMN on low-performing subjects. Finally, we study the use of CMMN in conjunction with domain adaptation approaches. In order to promote research reproducibility, code is available on github [2], and the datasets used are publicly available.

### 4.1 Experimental setup

**Sleep staging datasets**   We use three publicly available datasets: Physionet (a.k.a SleepEDF) [3], SHHS [4, 37] and MASS [2]. On all datasets, we want to perform sleep staging from 2-channels EEG signals. The considered EEG channels are 2 bipolar channels, Fpz-Cz and Pz-Cz that have been known to provide discriminant information for sleep staging. Note that those channels were not available on the SHHS dataset, and we used the C3-A2 and C4-A1 instead. This brings another level of variability in the data. More details about the datasets are available as supplementary material.

**Pre-processing**   For all experiments, we keep 60 subjects of each dataset and the two EEG channels. The same pre-processing is applied to all sensors. First, the recordings are low-pass filtered with a 30 Hz cutoff frequency, then the signals are resampled to 100 Hz. Then we extract 30 s epochs having a unique class. This pre-processing is common in the field [6, 38]. All the data extraction and the pre-processing steps are done with MNE-BIDS [39] and MNE-Python [40].

**Neural network architectures and training**   Many neural network architectures dedicated to sleep staging have been proposed [8, 7, 41]. In the following, we choose to focus on two architectures: Chambon [6] and DeepSleepNet [7]. For both architectures, we use the implementation from braindecode [42]. Chambon is an end-to-end neural network proposed to deal with multivariate time series and is composed of two convolutional layers with non-linear activation functions. DeepSleepNet is a more complex model with convolutional layers, non-linear activation functions, and a Bi-LSTM to model temporal sequences.

We use the Adam optimizer with a learning rate of $10^{-3}$ for Chambon and $10^{-4}$ with a weight decay of $1 \times 10^{-3}$ for DeepSleepNet. The batch size is set to 128 and the early stopping is done on a validation set corresponding to 20% of the subjects in the training set with a patience of 10 epochs. For all methods, we optimize the cross entropy with class weight, which amounts to optimizing for the balanced accuracy (BACC).

Various metrics are commonly used in the field such as Cohen's kappa, F1-score, or Balanced Accuracy (BACC) [36, 6]. We report here the BACC score as it is a metric well adapted to unbalanced classification tasks such as sleep staging. We also report in some experiments the gain the balanced accuracy, when using CMMN, of the 20% worst performing domains/subjects in the target domain denoted as $\Delta$BACC@20 in the following.

---

[2]https://github.com/PythonOT/convolutional-monge-mapping-normalization

| Datasets \Norm. | None [7] | Sample [6] | Session [5] | CMMN |
|---|---|---|---|---|
| MASS→MASS | 73.9 ± 1.4 | 75.1 ± 1.0 | 76.0 ± 2.4 | **76.2 ± 2.2** |
| Phys.→Phys. | 68.8 ± 2.8 | 69.2 ± 2.7 | 69.4 ± 3.0 | **71.7 ± 2.4** |
| SHHS→SHHS | 55.1 ± 12.5 | 61.2 ± 3.8 | 60.8 ± 2.6 | **64.3 ± 2.7** |
| MASS→Phys. | 55.9 ± 3.1 | 58.4 ± 2.4 | 57.5 ± 2.0 | **62.3 ± 1.5** |
| MASS→SHHS | 45.8 ± 3.3 | 41.8 ± 3.6 | 37.4 ± 3.6 | **47.6 ± 4.0** |
| Phys.→MASS | 63.8 ± 3.9 | 64.0 ± 2.7 | 63.7 ± 2.3 | **68.3 ± 2.5** |
| Phys.→SHHS | **53.9 ± 3.2** | 45.6 ± 2.1 | 47.9 ± 1.8 | 51.6 ± 1.8 |
| SHHS→MASS | 48.7 ± 4.8 | 57.0 ± 2.8 | 51.8 ± 6.4 | **64.5 ± 2.8** |
| SHHS→Phys. | 52.6 ± 4.2 | 55.0 ± 2.7 | 52.4 ± 4.1 | **58.3 ± 1.7** |
| Mean | 57.6 ± 4.3 | 58.6 ± 2.6 | 57.4 ± 3.1 | **62.7 ± 2.4** |

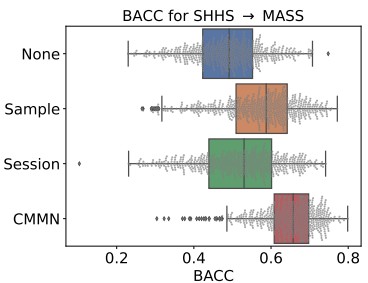

Table 1: Balanced accuracy (BACC) for different normalizations and different train/test dataset pairs (left). Boxplot for all normalization approaches on the specific pair SHHS→MASS (right). CMMN outperforms other normalizations.

**Filter size and sensitivity analysis for** `CMMN`  Our method has a unique hyperparameter that is the size of the filter $F$ used. In our experiments, we observed that, while this parameter has some impact, it is not critical and has quite a wide range of values ($[8, 512]$) that leads to systematic performance gains. We provide in supplementary material a sensitivity analysis of the performance for different adaptation scenarios (pairs of datasets). It shows that the value $F = 128$ is a good trade-off that we used below for all experiments.

## 4.2 Comparison between different normalizations

We now evaluate the ability of `CMMN` to adapt to new subjects within and across datasets and also between two sessions of the same subject.

**Classical normalizations**  We first compare `CMMN` to several classical normalization strategies. We compare the use of raw data [7] letting the neural network learn the normalization from the data (`None`), standard normalization of each 30 s samples [6] that discard global trend along the session (`Sample`) and finally normalization by session that consists in our case to perform normalization independently on each domain (`Session`) [5]. We train the `Chambon` neural network on the source data of one dataset and evaluate on the target data of all other datasets for different splits.
The BACC for all dataset pairs and normalization are presented in the Table 1. The three classical normalizations have similar performances with a slight edge for `Sample` on average. All those approaches are outperformed by `CMMN` in 8 out of 9 dataset pairs with an average gain of 4% w.r.t. the best performing `Sample`. This is also visible on the Boxplot on the right where the BACC for all subjects/domains (*i.e.,* points) is higher for `CMMN`. Since the very simple `Sample` normalization is the best-performing competitor, we used it as a baseline called `No Adapt` in the following.

**Domain specific normalization**  We have shown above that the `CMMN` approach allows to better cope with distribution shifts than standard data normalizations. This might be explained by the fact that `CMMN` normalization is domain/subject specific. This is why we now compare to several existing domain-specific normalizations. To do this we adapt the method of Liu *et al.* [11] which was designed for image classification. We implemented domain-specific convolution layers (`Conv`), batch normalization (`Norm`), or both (`ConvNorm`). In practice, we have one layer per domain that is trained (jointly with the predictor $f$) only on data from the corresponding domain. The limit of domain-specific normalization is that all test domains must be represented in the training set. Otherwise, if a new domain arrives in the test set, no layer specific to that domain will have been trained.
To be able to compare these methods to `CMMN`, we use for this section the Physionet dataset for which two sessions are available for some subjects. The first sessions are considered as the training set where the domains are the subjects and the second sessions are split between the validation set (20%) and the test set (80%). The validation set is used to do the early stopping, and validate the kernel size of the subject-specific convolution for `Conv` and `ConvNorm`.
We can see in Table 2 that for cross-session adaptation, the gain with `CMMN` is smaller than previous results (1% BACC gain), which can be explained by the presence of subjects data in both domains resulting in a smaller shift between distribution. However, `CMMN` outperforms all other subject-specific normalizations which are struggling to improve the results (*i.e.,* around 4% of BACC loss).

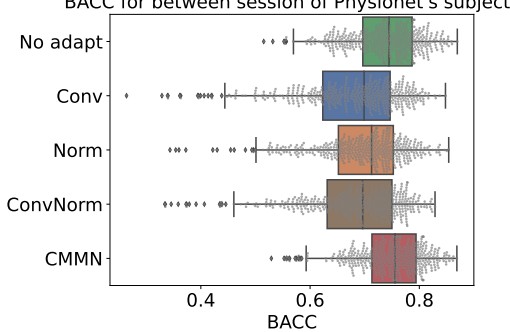

| Normalization | BACC |
|---|---|
| `No Adapt` | $73.7 \pm 0.7$ |
| `Conv` [11] | $67.5 \pm 2.7$ |
| `Norm` [11] | $69.4 \pm 1.6$ |
| `ConvNorm` [11] | $68.1 \pm 1.3$ |
| `CMMN` | $\mathbf{74.8 \pm 0.6}$ |

Table 2: Balanced accuracy (BACC) for different subject-specific normalizations and `CMMN` (left). Boxplot for all normalization approaches (right). CMNN outperforms subject-specific normalizations.

| Architecture | Chambon [6] | | DeepSleepNet [7] | |
|---|---|---|---|---|
| | No Adapt | CMMN | No Adapt | CMMN |
| MASS→MASS | $75.1 \pm 1.0$ | $\mathbf{76.2 \pm 2.2}$ | $\mathbf{73.3 \pm 1.7}$ | $73.1 \pm 2.6$ |
| Phys.→Phys. | $69.2 \pm 2.7$ | $\mathbf{71.7 \pm 2.4}$ | $66.5 \pm 2.5$ | $\mathbf{69.4 \pm 2.5}$ |
| SHHS→SHHS | $61.2 \pm 3.8$ | $\mathbf{64.3 \pm 2.7}$ | $58.7 \pm 2.3$ | $\mathbf{60.1 \pm 3.5}$ |
| MASS→Phys. | $58.4 \pm 2.4$ | $\mathbf{62.3 \pm 1.5}$ | $50.1 \pm 2.4$ | $\mathbf{54.5 \pm 1.2}$ |
| MASS→SHHS | $41.8 \pm 3.6$ | $\mathbf{47.6 \pm 4.0}$ | $38.3 \pm 2.6$ | $\mathbf{47.8 \pm 2.4}$ |
| Phys.→MASS | $64.0 \pm 2.7$ | $\mathbf{68.3 \pm 2.5}$ | $59.5 \pm 1.0$ | $\mathbf{62.1 \pm 1.9}$ |
| Phys.→SHHS | $45.6 \pm 2.1$ | $\mathbf{51.6 \pm 1.8}$ | $45.2 \pm 2.2$ | $\mathbf{48.6 \pm 1.6}$ |
| SHHS→MASS | $57.0 \pm 2.8$ | $\mathbf{64.5 \pm 2.8}$ | $51.2 \pm 5.9$ | $\mathbf{56.8 \pm 6.1}$ |
| SHHS→Phys. | $55.0 \pm 2.7$ | $\mathbf{58.3 \pm 1.7}$ | $48.6 \pm 5.8$ | $\mathbf{54.7 \pm 6.8}$ |
| Mean | $58.6 \pm 2.6$ | $\mathbf{62.7 \pm 2.4}$ | $54.6 \pm 2.9$ | $\mathbf{58.6 \pm 3.2}$ |

Table 3: Balanced accuracy (BACC) for different train/test dataset pairs and for different architectures (`Chambon`/`DeepSleepNet`). CMNN works independently of the network architecture.

### 4.3 Study of the performance gain: neural architecture and human subjects

Previous experiments have shown the superiority of `CMMN` over all the other normalizations. In this section, we study the behavior of `CMMN` on different neural network architectures and study which subject gains the most performance gain.

**Performance of `CMMN` with different architectures**  In addition to `Chambon` that was used in the previous experiments, we now evaluate `CMMN` considering a different network architecture: `DeepSleepNet` [7]. The results for both architectures are reported in Table 3, where `CMMN` is consistently better for both architectures. Notably, the only configuration where the gain is limited is MASS→MASS with `DeepSleepNet` because MASS is the easiest dataset with less variability than other pairs. Finally, we were surprised to see that `DeepSleepNet` does not perform as well as `Chambon` on cross-dataset adaptation, probably due to overfitting caused by a more complex architecture.

**Performance gain on low-performing subjects**  In medical applications, it is often more critical to have a model that has a low failure mode, rather than the best average accuracy. As a first step toward studying this, we report two scatter plots reported in Table 4 plotting the BACC for individual target subjects without adaptation as a function of the BACC with `CMMN`, for different architectures and dataset pairs. First, the majority of the subjects are above the axis $x = y$, which means that `CMMN` improves their score. But the most interesting finding is the large improvement for the low-performing subjects that can gain from 0.3 to 0.65 BACC.

We also provide in Table 4 the $\Delta$BACC@20, that is the average BACC gain on the 20% lowest performing subjects without adaptation. On average, both architectures increase by 7% the BACC on those subjects, when it is only increased by 4% for all subjects. Some $\Delta$BACC@20 are even greater

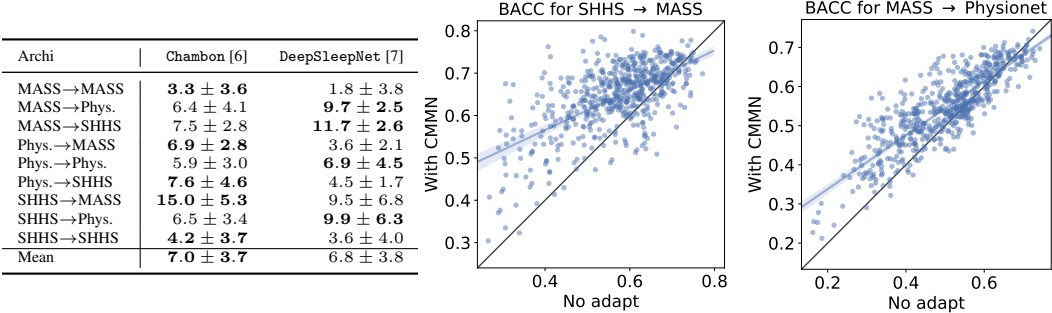

| Archi | Chambon [6] | DeepSleepNet [7] |
|---|---|---|
| MASS→MASS | **3.3 ± 3.6** | 1.8 ± 3.8 |
| MASS→Phys. | 6.4 ± 4.1 | **9.7 ± 2.5** |
| MASS→SHHS | 7.5 ± 2.8 | **11.7 ± 2.6** |
| Phys.→MASS | **6.9 ± 2.8** | 3.6 ± 2.1 |
| Phys.→Phys. | 5.9 ± 3.0 | **6.9 ± 4.5** |
| Phys.→SHHS | **7.6 ± 4.6** | 4.5 ± 1.7 |
| SHHS→MASS | **15.0 ± 5.3** | 9.5 ± 6.8 |
| SHHS→Phys. | 6.5 ± 3.4 | **9.9 ± 6.3** |
| SHHS→SHHS | **4.2 ± 3.7** | 3.6 ± 4.0 |
| Mean | **7.0 ± 3.7** | 6.8 ± 3.8 |

Table 4: $\Delta$BACC@20 for different train/test dataset pairs and for different architectures (Chambon/DeepSleepNet) (left). Scatter plot of balanced accuracy (BACC) with No Adapt as a fuction of BACC with CMMN and the dataset pair SHHS → MASS with Chambon (center) and on the dataset pair MASS → Physionet with DeepSleepNet (right). CMMN leads to a big performance boost on the low-performing subjects.

|  | BACC | | | | $\Delta$BACC@20 | | |
|---|---|---|---|---|---|---|---|
| Adapt | No Adapt | DANN | CMMN | CMMN+DANN | DANN | CMMN | CMMN+DANN |
| MASS->Phys. | 59.2 ± 6.1 | 60.9 ± 1.0 | 62.9 ± 0.8 | **62.9 ± 1.2** | 2.0 ± 5.9 | 5.3 ± 5.8 | **5.5 ± 6.2** |
| MASS->SHHS | 44.5 ± 6.0 | 44.3 ± 2.1 | **50.3 ± 4.2** | 49.9 ± 1.9 | 4.2 ± 7.1 | 7.6 ± 2.9 | **10.4 ± 5.9** |
| Phys.->MASS | 65.3 ± 1.4 | 65.2 ± 0.7 | 68.9 ± 1.0 | **69.1 ± 1.0** | 0.4 ± 1.3 | **6.3 ± 2.0** | 5.6 ± 2.7 |
| Phys.->SHHS | 43.1 ± 6.3 | 45.4 ± 2.6 | **50.0 ± 4.3** | 49.9 ± 2.4 | 2.8 ± 5.3 | 8.2 ± 5.7 | **9.7 ± 6.1** |
| SHHS->MASS | 59.9 ± 3.4 | 59.4 ± 1.1 | **66.5 ± 2.5** | 66.3 ± 0.9 | 0.4 ± 2.8 | **13.4 ± 3.0** | 12.6 ± 2.7 |
| SHHS->Phys. | 57.1 ± 3.9 | 57.2 ± 2.2 | **61.6 ± 3.1** | 59.4 ± 2.6 | 4.1 ± 3.7 | 10.0 ± 6.9 | **10.9 ± 5.7** |
| Mean | 54.9 ± 4.5 | 55.4 ± 1.6 | **60.0 ± 2.7** | 59.6 ± 1.7 | 2.3 ± 4.4 | 8.5 ± 4.4 | **9.1 ± 4.9** |

Table 5: Balanced accuracy (BACC) and $\Delta$BACC@20 for different train/test dataset pairs and for different adaptation methods. CMMN outperforms DANN and CMMN+DANN on average on all subjects. Combining CMMN DANN improves the lower-performing subjects.

than 10% on some dataset pairs. These results show the consistency of the method on all subjects but also the huge impact on the more challenging ones.

## 4.4 Complementarity of CMMN with Domain Adaptation

We have shown in the previous experiments that CMMN is clearly the best normalization in many settings. But the main idea of CMMN is to adapt the raw signals to a common barycentric domain. Interestingly, many Domain Adaptation (DA) methods also try to reduce the discrepancies between datasets by learning a feature representation that is invariant to the domain. In this section, we compare the two strategies and investigate if they are complementary.

We implement the celebrated DA method DANN [18] that aims at learning a feature representation that is invariant to the domain using an adversarial formulation. Note that this DA approach is much more complex than CMMN because it requires to have access to the target data during training and a model needs to be trained for each new target domain. The choice of hyperparameters for DA methods is not trivial in the absence of target labels. But since we have access to several target domains, we propose to select the weight parameters for DANN using a validation on 20% of the target subjects [43]. Note that this is not a realistic setting since in real applications the target domains are usually not labeled, yet it is a way to compare the two approaches in a configuration favorable for DA. We focus on cross-dataset adaptation where many shifts are known to exist: different sensors (SHHS vs Physionet/MASS), doctor scoring criteria (SHHS/MASS vs Physionet), or brain activity (SHHS vs MASS vs Physionet).

We report in Table 5 the BACC and $\Delta$BACC@20 for all dataset pairs and all combinations of CMMN and DANN with Chambon. First, we can see that the best approaches are clearly CMMN and CMMN+DANN. CMMN is better in BACC on 4/6 dataset pairs and CMMN+DANN is better in $\Delta$BACC@20 on 4/6 dataset pairs. First, it is a very impressive performance for CMMN that is much simpler than DANN and again does not use target data when learning the predictor $f$. But it also illustrates the interest of CMMN+DANN especially for low-performing subjects.

Table 6: BACC for different PSD targets for the mapping.

| Target PSD | None | Barycenter | Powerlaw | Whitening |
|---|---|---|---|---|
| MASS→MASS | $75.1 \pm 1.0$ | $\mathbf{77.1 \pm 1.3}$ | $75.6 \pm 2.4$ | $73.2 \pm 6.4$ |
| MASS→Physionet | $58.4 \pm 2.4$ | $62.6 \pm 2.0$ | $63.1 \pm 1.5$ | $\mathbf{63.2 \pm 1.0}$ |
| MASS→SHHS | $41.8 \pm 3.6$ | $50.4 \pm 8.0$ | $51.9 \pm 3.0$ | $\mathbf{52.4 \pm 2.4}$ |
| Physionet→MASS | $64.0 \pm 2.7$ | $\mathbf{68.0 \pm 1.5}$ | $66.7 \pm 2.5$ | $65.9 \pm 2.6$ |
| Physionet→Physionet | $69.2 \pm 2.7$ | $\mathbf{72.3 \pm 1.8}$ | $66.9 \pm 16.5$ | $71.3 \pm 1.9$ |
| Physionet→SHHS | $45.6 \pm 2.1$ | $52.5 \pm 1.9$ | $53.9 \pm 1.7$ | $\mathbf{54.0 \pm 2.4}$ |
| SHHS→MASS | $57.0 \pm 2.8$ | $\mathbf{65.9 \pm 3.2}$ | $59.0 \pm 13.9$ | $59.0 \pm 13.8$ |
| SHHS→Physionet | $55.0 \pm 2.7$ | $\mathbf{62.0 \pm 3.2}$ | $56.5 \pm 13.1$ | $55.9 \pm 12.8$ |
| SHHS→SHHS | $61.2 \pm 3.8$ | $63.5 \pm 3.2$ | $\mathbf{63.6 \pm 2.6}$ | $62.4 \pm 2.8$ |
| Mean | $58.6 \pm 2.6$ | $\mathbf{63.8 \pm 2.9}$ | $61.9 \pm 6.4$ | $61.9 \pm 5.1$ |

## 4.5 Different PSD targets

CMMN has shown a significant boost in performance for different adaptations in various settings. The method consists of learning a barycenter, and mapping all domains to this barycenter to obtain a homogeneous frequency spectrum. Mapping to the barycenter is one way to achieve this, but it is reasonable to evaluate the optimality of this choice by comparing to alternatives. For example, the classical method called spectral whitening is equivalent to OT mapping towards a uniform PSD, which is equivalent to a white noise PSD. In this part, we propose to compare the mapping to different PSD targets: barycenter (classic CMMN), white noise PSD (whitening), or a non-uniform power-law. The last PSD target is a mathematical distribution that describes a functional relationship between frequency and magnitude, where magnitude is inversely proportional to the power of the frequency $P(f) = af^{a-1}$. Below we selected $a = 0.659$ for the experiment.

The table Table 6 gives the BACC score of the mapping for the different PSD targets and shows the importance of the reference PSD. First, we can see that mapping to a PSD increases the score significantly w.r.t. to raw data. Second, the mapping to the Wasserstein barycenter (*i.e.,* CMMN) is not always the better performer (only 5/9), but overall, CMMN gives better results (2% higher) and with less variance. The robustness of the chosen PSD target, coupled with the fast computation of the barycenter, makes CMMN a strong and easy-to-compute normalization method for time series.

## 5 Conclusion

We proposed in this paper a novel approach for the normalization of bio-signals that can adapt to the spectral specificities of each domain while being a test-time adaptation method that does not require retraining a new model. The method builds on a new closed-form solution for computing Wasserstein barycenters on stationary Gaussian random signals. We showed that this method leads to a systematic performance gain on different configurations of data shift (between subjects, between sessions, and between datasets) and on different architectures. We also show that CMMN benefits greatly the subjects that had bad performances when trained jointly without sacrificing performance on the well-predicted subjects. Finally, we show that CMMN even outperforms DA methods and can be used in conjunction with DA for even better results.

Future work will investigate the use of CMMN for other biomedical applications and study the use of the estimated filters $\mathbf{h}_k$ as vector representations of the subjects that can be used for interpretability. Finally, we believe that a research direction worth investigating is the federated estimation of CMMN with the objective of learning an unbiased estimator in the context of differential privacy [44, 45].

## 6   Acknowledgement

The authors thank Antoine Collas, Cédric Allain, and Nicolas Courty for their valuable comments on the manuscript, and Lina Dalibard for her help with Figure 1. Numerical computation was enabled by the scientific Python ecosystem: NumPy [46], SciPy [47], Matplotlib [48], Seaborn [49], PyTorch [50], and MNE for EEG data processing [40]. This work was partly supported by the grants ANR-20-CHIA-0016 and ANR-20-IADJ-0002 and ANR-23-ERCC-0006-01 from Agence nationale de la recherche (ANR).

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

# A  Supplementary material

## A.1  Proof of the convolutional Wasserstein barycenter

PROOF.  Consider $K$ centered stationary Gaussian signals of covariance $\boldsymbol{\Sigma}_k$ $\mathbf{p}_k$ (respectively PSD $\mathbf{p}_k$) with $k \in [K]$, the Wasserstein barycenter of the $K$ signals is a centered stationary Gaussian signal of PSD $\bar{\mathbf{p}}$ with:

$$\bar{\boldsymbol{\Sigma}} = \frac{1}{K} \sum_{k=1}^{K} \left( \bar{\boldsymbol{\Sigma}}^{\frac{1}{2}} \boldsymbol{\Sigma}_k \bar{\boldsymbol{\Sigma}}^{\frac{1}{2}} \right)^{\frac{1}{2}} \tag{8}$$

The signals are supposed to be stationary. Therefore the covariance matrix is a Toeplitz circulant matrix. The circulant matrix can be diagonalized by the Discrete Fourier Transform (DFT) $\boldsymbol{\Sigma} = \mathbf{F}\text{diag}(\mathbf{p})\mathbf{F}^*$, with $\mathbf{F}$ and $\mathbf{F}^*$ the Fourier transform operator and its inverse, and $\mathbf{p}$ the Power Spectral Density (PSD) of the signal. The above equation becomes:

$$\bar{\mathbf{p}} = \frac{1}{K} \sum_{k=1}^{K} \left( \bar{\mathbf{p}}^{\odot \frac{1}{2}} \odot \mathbf{p}_k \odot \bar{\mathbf{p}}^{\odot \frac{1}{2}} \right)^{\odot \frac{1}{2}} \tag{9}$$

The matrix square root and the inverse become element-wise square root and inverse. The equation becomes easier, and the term can be managed to isolate $\bar{\mathbf{p}}$:

$$\bar{\mathbf{p}} = \frac{1}{K} \sum_{k=1}^{K} \left( \bar{\mathbf{p}}^{\odot \frac{1}{2}} \odot \mathbf{p}_k \odot \bar{\mathbf{p}}^{\odot \frac{1}{2}} \right)^{\odot \frac{1}{2}}$$

$$\bar{\mathbf{p}} = \frac{1}{K} \sum_{k=1}^{K} \bar{\mathbf{p}}^{\odot \frac{1}{2}} \odot \mathbf{p}_k^{\odot \frac{1}{2}}$$

$$\bar{\mathbf{p}}^{\odot \frac{1}{2}} = \frac{1}{K} \sum_{k=1}^{K} \mathbf{p}_k^{\odot \frac{1}{2}}$$

$$\bar{\mathbf{p}} = \left( \frac{1}{K} \sum_{k=1}^{K} \mathbf{p}_k^{\odot \frac{1}{2}} \right)^{\odot 2}$$

PROOF.  A second possible proof is considering the optimization problem *w.r.t* $\boldsymbol{\Sigma}$:

$$\bar{\boldsymbol{\Sigma}} = \arg\min_{\boldsymbol{\Sigma}} \sum_{k=1}^{K} \text{Tr} \left( \boldsymbol{\Sigma} + \boldsymbol{\Sigma}_k - 2 \left( \boldsymbol{\Sigma}^{\frac{1}{2}} \boldsymbol{\Sigma}_k \boldsymbol{\Sigma}^{\frac{1}{2}} \right)^{\frac{1}{2}} \right) . \tag{10}$$

As mentioned above, it is possible to use the PSD $\mathbf{p}$ to transform the equation into an element-wise problem as before:

$$\bar{\mathbf{p}} = \arg\min_{\mathbf{p}} \sum_{k=1}^{K} \| \mathbf{p} + \mathbf{p}_k - 2 \left( \mathbf{p}^{\frac{1}{2}} \odot \mathbf{p}_k \odot \mathbf{p}^{\odot \frac{1}{2}} \right)^{\odot \frac{1}{2}} \|_1$$

$$\bar{\mathbf{p}} = \arg\min_{\mathbf{p}} \sum_{k=1}^{K} \| \mathbf{p} + \mathbf{p}_k - 2 \mathbf{p}^{\odot \frac{1}{2}} \odot \mathbf{p}_k^{\odot \frac{1}{2}} \|_1 \tag{11}$$

After derivation, the $\bar{\mathbf{p}}$ minimizing the optimization problem is given by:

$$\bar{\mathbf{p}} = \left( \frac{1}{K} \sum_{k=1}^{K} \mathbf{p}_k^{\odot \frac{1}{2}} \right)^{\odot 2} \tag{12}$$

## A.2 Computation

The training is done on Tesla V100-DGXS-32GB with Pytorch. We are considering the following train settings: `Chambon` architecture with a learning rate of $1e^{-3}$ for Adam optimizer and a patience of 10 for the early stopping. The training for one dataset pair with ten different splits and seeds lasts approximately 1 hour. The data processing time with the CMMN is insignificant compared to the network's computation time (a few minutes).

## A.3 Dataset descriptions

**SHHS**  The Sleep Heart Health Study is a multi-center cohort study proposed by the National Heart Lung & Blood Institute [4, 37] to help detect cardiovascular disease and sleep disorders. This large dataset comprises 6441 subjects (age $63.1 \pm 11.2$) from 1995 to 1998. Five sensors are available for each subject: 2 EEGs from C3-A2 and C4-A1 channels, left and right EOGs, and one EMG. The EEGs have a sampling rate of 125 Hz. The hypnograms were scored according to the Rechtschaffen and Kales criteria [51].

**MASS**  The Montreal Archive of Sleep Studies comprised five different subsets of recordings. This paper focuses on the SS3 with recordings from 62 healthy subjects (age $42.5 \pm 18.9$). For each subject, 20 EEGs, left and right EOGs, and 3 EMGs are available. We reduced the number of EEG channels to 2 bipolar channels, Fpz-Cz and Pz-Cz, obtained by montage reformatting. The EEGs have a sampling rate of 256 Hz. The MASS hypnograms were scored according to the AASM criteria [52].

**Physionet SleepEDF**  This dataset comprises two subsets, one for the age effect in healthy subjects (SC) and one for the Temazepam effect on sleep (ST). We focused on the SC subset where 78 subjects are available (age $28.7 \pm 2.9$). Each recording comprises 2 EEGs from Fpz-Cz and Pz-Cz channels, 1 EOG, and 1 EMG. The EEGs have a sampling rate of 100 Hz. Some of the subjects have two sessions of PSGs available. The hypnograms were scored according to the Rechtschaffen and Kales criteria [51]. The stages N3 and N4 have been merged for the following.

**Ethical consideration**  All datasets used in our experiments are anonymized and public datasets that have already passed an ethics committee before recording.

## A.4 Sensitivity analysis to the filter size

Several adaptations across different dataset pairs are done to compare the effect of the filter size. The smallest filter size means no transformation, and the largest size corresponds to a perfect transformation between the two signals. For each parameter, ten training are done over data from the source dataset and then tested over data from the target dataset.

To evaluate the benefit of the method, we measure the $\Delta$BACC corresponding to the difference between the balanced accuracy score with monge mapping and the balanced accuracy score without monge mapping (*i.e.,* using `Sample` normalization). The Figure 2 shows the evaluation of $\Delta$BACC with filter size for different dataset pairs. The slightest improvement is for adaptation between the same dataset, which is logical because there is less difference to compensate between the subjects. And the best improvement is for the most challenging task, adaptation between datasets with different sensors (MASS/Physionet $\rightarrow$ SHHS). The mapping did not capture enough information for the smallest filter size to reduce the difference between distributions. On the other hand, the bigger the distribution gap between datasets, the bigger the filter size helps to adapt. Indeed, for an adaptation between the same dataset, having a filter size close to the sample size decreases the performance (see MASS $\rightarrow$ MASS, Physionet $\rightarrow$ Physionet), while for an adaptation between two different or very different datasets, increasing the filter size causes the performance to remain the same (MASS $\leftrightarrow$ Physionet) or even increases the scores considerably (MASS/Physionet $\rightarrow$ SHHS).

## A.5 Boxplot of BACC for different data normalizations, different architectures, and different dataset pairs

As shown in the experimental section of the paper, CMMN outperforms standard normalization (`Session` or `Sample`) for different architectures. Here we provide more boxplots for other dataset

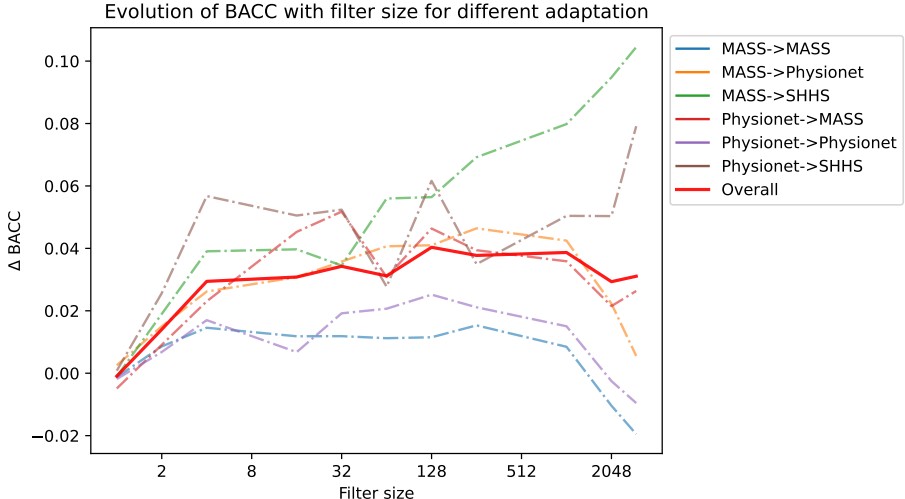

Figure 2: Evolution of the ΔBACC (BACC - BACC without mapping) for different filter sizes for different adaptation problems

pairs than in the main article. For `Chambon`, the figure 3f shows again that `CMMN` outperforms other normalizations except with dataset SHHS in target. Indeed, if `CMMN` is still better than `Sample` and `Session`, using no normalization is better for this experimental setting. SHHS is the most different dataset since the sensors used differ from Physionet and MASS, which can explain this difference. Even when SHHS is in train, `CMMN` is better than `None`.

The results for `DeepSleepNet` in figure 4f are slightly different. If `CMMN` is still the better performer overall, the best standard normalization is `None`. Using no normalization is better in 5/6 dataset pairs over `Sample` and `Session`. These results were expected since no normalization was used in the paper proposing `DeepSleepNet` [7].

For the sake of simplicity, we used `Sample` normalization before `CMMN` and also for the baseline `No Adapt.` even for `DeepSleepNet`.

## A.6   Scatterplot for different architectures and different dataset pairs

One significant benefit of `CMMN` is to have a massive impact on low-performing subjects. The scatter plots of balanced accuracy (BACC) with `No Adapt` as a function of BACC with `CMMN` emphasize this effect. In this section, the plots in figure 5f and figure 6f show the increase in all subjects. For both architectures, the axis of the linear regression is always above the axis $x = y$. The differences between the two axes are generally higher for lower performances, which means that the worst the performance is, the higher the increase is. For the eccentric dots on the left (*i.e.,* low-performing subjects), the increase is generally around 8% (see Table 4). When SHHS is the training set, low-performing subjects get a considerable boost. On the other hand, for Physionet $\leftrightarrow$ MASS with `Chambon`, some subjects lose accuracy, but it remains rare compared to the average gain on all subjects.

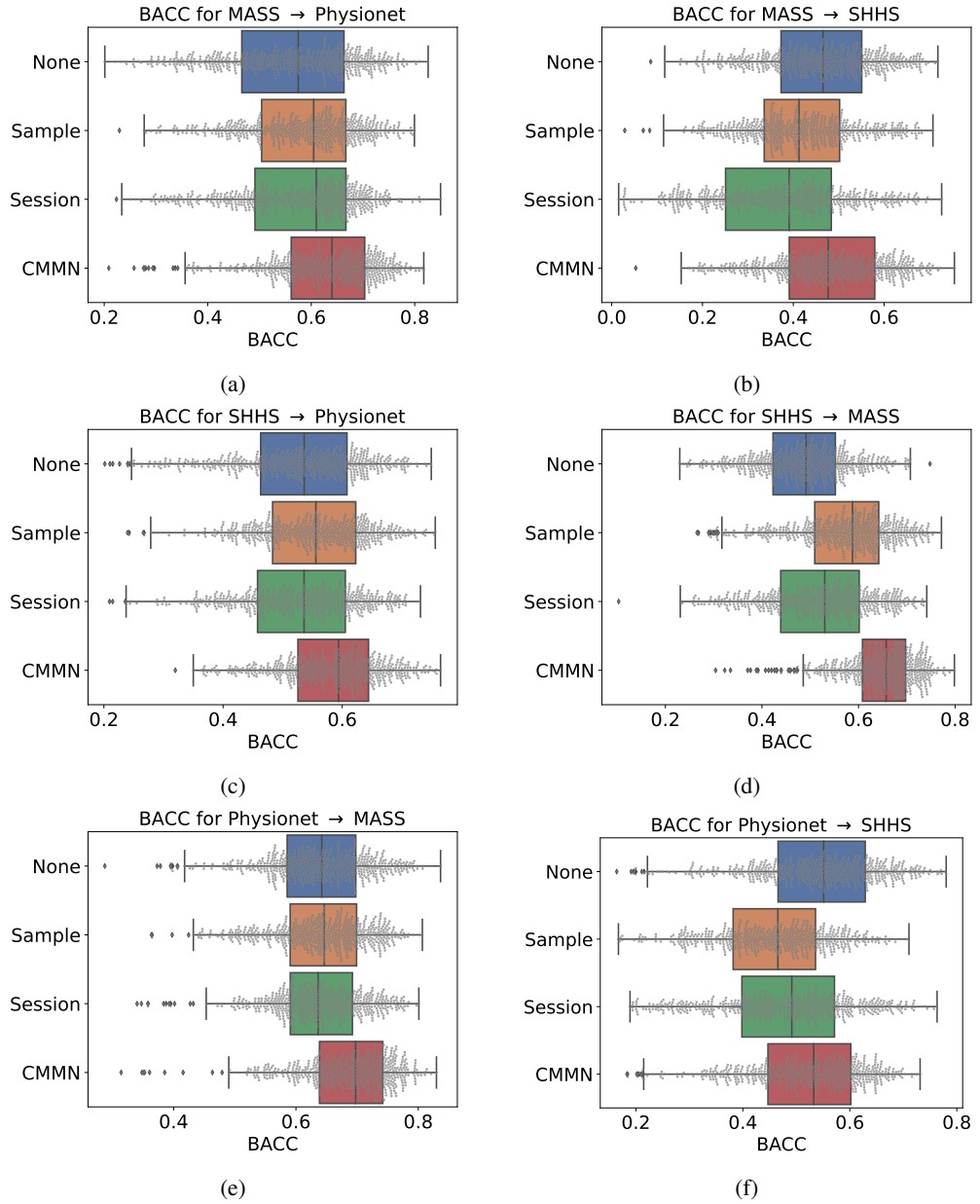

Figure 3: Boxplot of balanced accuracy (BACC) for different normalizations and different train/test dataset pairs with `Chambon`.

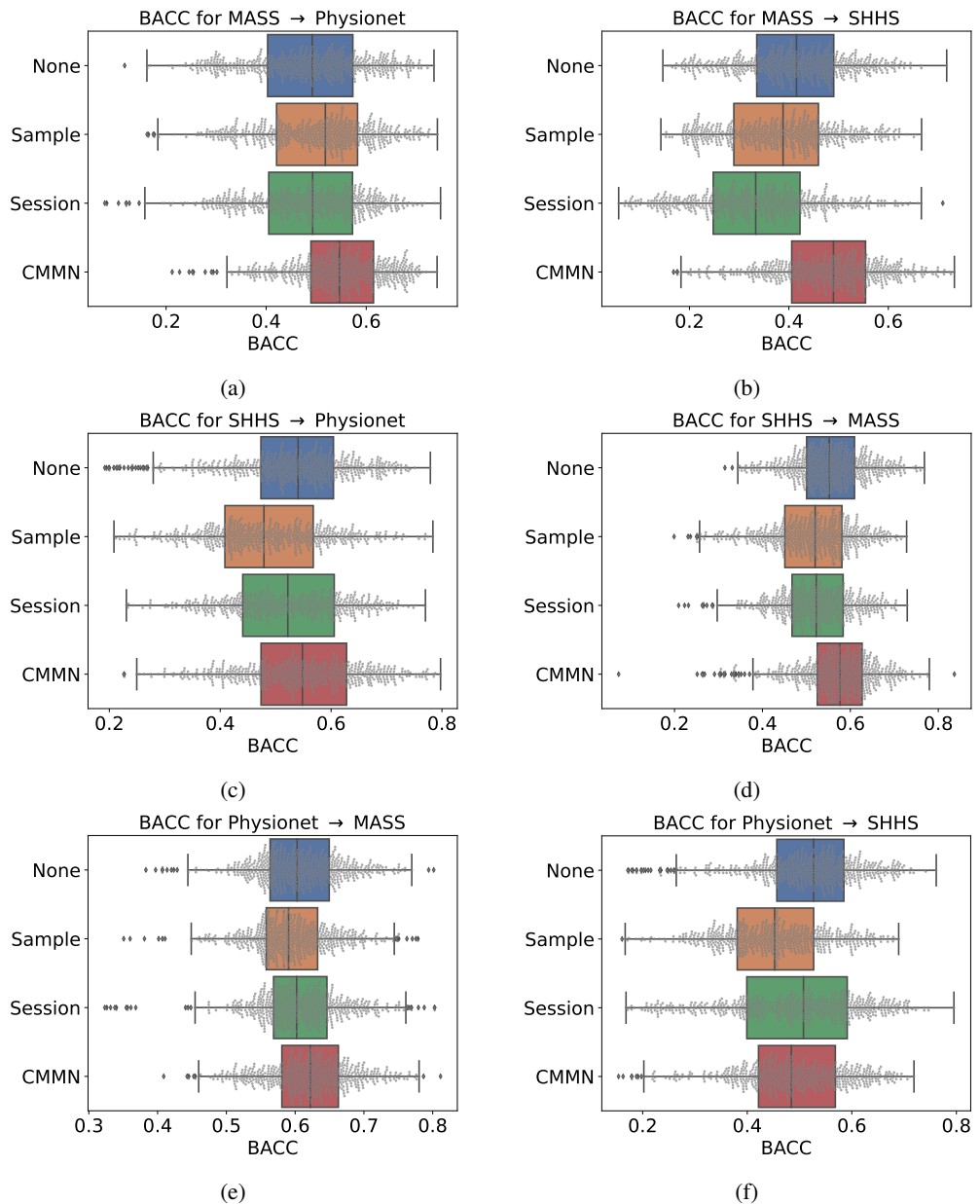

Figure 4: Boxplot of balanced accuracy (BACC) for different normalizations and different train/test dataset pairs with `DeepSleepNet`.

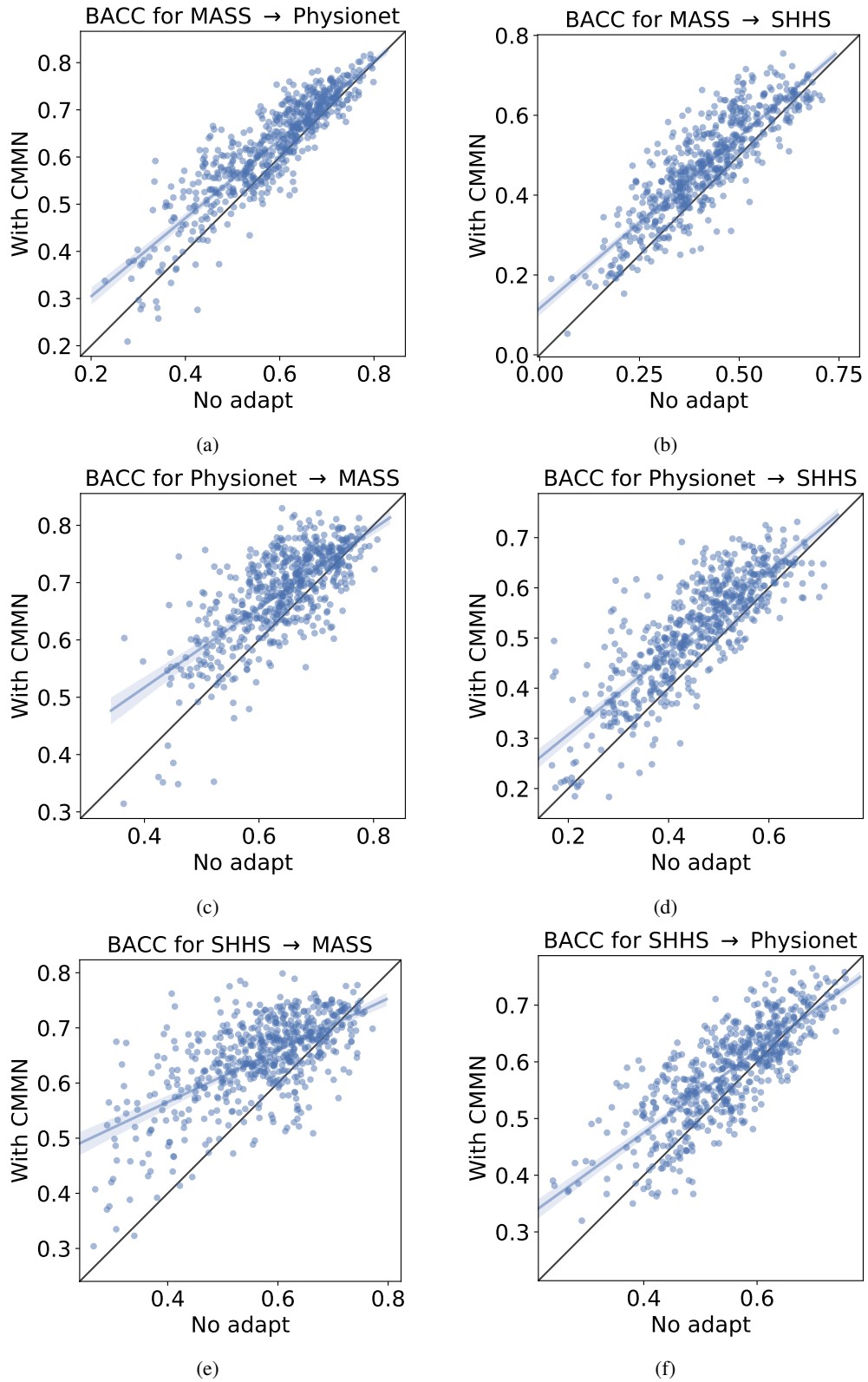

Figure 5: Scatter plot of balanced accuracy (BACC) with `No Adapt` as a function of BACC with `CMMN` for different dataset pairs with `Chambon`.

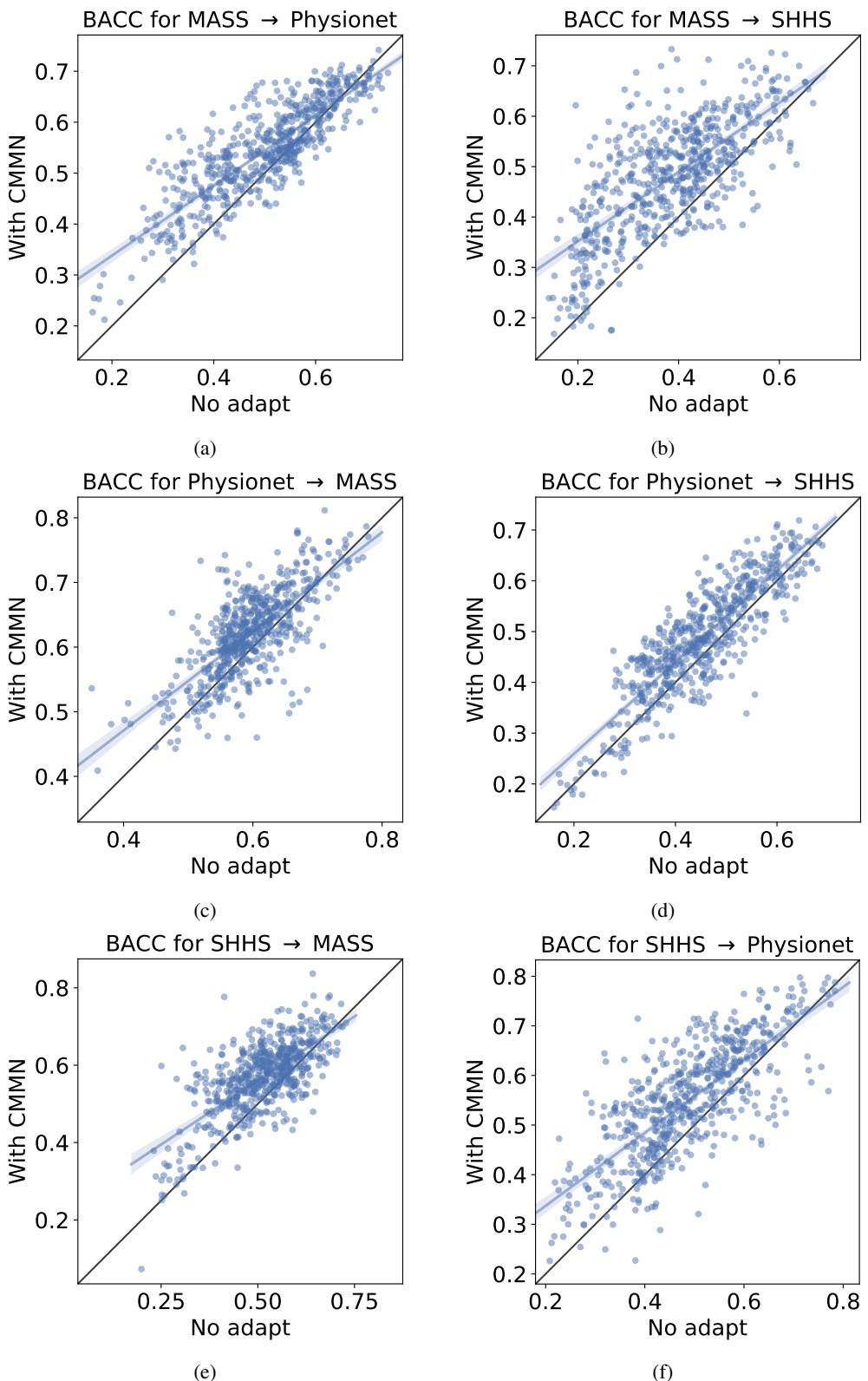

Figure 6: Scatter plot of balanced accuracy (BACC) with `No Adapt` as a function of BACC with CMMN for different dataset pairs with `DeepSleepNet`

