# OpenReview forum: "Convolution Monge Mapping Normalization for learning on sleep data"
_NeurIPS.cc/2023/Conference — NeurIPS 2023 poster_

### Official Review · Reviewer_uQaR · 2023-07-04

**Soundness:** 4 excellent
**Presentation:** 4 excellent
**Contribution:** 3 good
**Rating:** 8
**Confidence:** 5

**Summary:**

The authors introduce an easily computable optimal transport method (Convolutional Monge Mapping Normalisation) for power spectral density data that can be assumed to originate from Gaussian and stationary data. These assumptions allows for a closed from analytical expression that uses the convolutions to reduce the data shifts caused by different measurement conditions.

The method is used for biological data (EEG) for classification of the stages of sleep, where the measurements are affected by individual physiological differences of the data subjects, different placements and number of the electrodes or different electrode impedances or even hardware used in the data acquisition.

The analytical normalisation method is compared with other normalisation schemes . The computational loads of different schemes is analysed . Different neural network architectures are studied to find out what are the benefits of the normalisation in practice - as well as the variation between different data sets.

**Strengths:**

The authors provide an excellent introduction of their low optimal transfer method. In comparisons their method is both simpler than others (in terms of the computational load) and perform better than the state of the art.

The manuscript studies thoroughly the benefits with respects to the architecture of the neural network, different available data sets showing advances in balanced accuracy.

**Weaknesses:**

I would have liked to have seen a discussion of the validity of the assumptions made in the derivation of the CMMN method. How far of the optimal transport CMMN is, if the data is not Gaussian (would kurtosis be a good enough indicator?), nor stationary? In independent component analysis (ICA) one is essentially searching for the non-Gaussian combination of the data to separating signals. ICA is extensively used in EEG signal analysis as the the signal is by its nature a summations of electric activations in brain.

This question could be analysed with synthetic data to introduce tests for generic data sets to pass in order to gain best benefits from CMMN. This would increase the impact and adoption of the normalisation method described in the submission.

**Questions:**

See the weakness part.

**Limitations:**

See the weakness part.

---

> ### Author Rebuttal · Authors · 2023-08-08
>
> > I would have liked to have seen a discussion of the validity of the assumptions made in the derivation of the CMMN method. How far of the optimal transport CMMN is, if the data is not Gaussian (would kurtosis be a good enough indicator?), nor stationary? In independent component analysis (ICA) one is essentially searching for the non-Gaussian combination of the data to separating signals. ICA is extensively used in EEG signal analysis as the the signal is by its nature a summations of electric activations in brain.
>
> This is a very interesting question. Note that if there exists a symmetric positive definite linear transformation between the two distributions, the Monge mapping will recover precisely the transformation even on non-gaussian data (Lemma 1 in [27], also see "Linear OT mapping estimation" 2D example in the POT toolbox documentation that maps between non-gaussian distributions with success). We align second-order moments so if the signal is a heavy tail with positive kurtosis, it will remain like this after CMMN, and the network learned on top can learn what to do with this.
>
> > This question could be analysed with synthetic data to introduce tests for generic data sets to pass in order to gain best benefits from CMMN. This would increase the impact and adoption of the normalisation method described in the submission.
>
> To measure the non-gaussianity of the data and its impact on the performance we provide a new plot of the classification performance as a function of the kurtosis. This is detailed in the global response.

---

> > ### Comment · Reviewer_uQaR · 2023-08-15
> >
> > Thank you for addressing my question on the non-Gaussian features. I retain my opinion that the authors have provided a very interesting method for "distilling" data sets to make them more effective in training.

---

### Official Review · Reviewer_Dgon · 2023-07-05

**Soundness:** 3 good
**Presentation:** 3 good
**Contribution:** 3 good
**Rating:** 6
**Confidence:** 3

**Summary:**

The paper introduces a new and efficient method to normalise biomedical signals, especially EEG. It experiments of an established benchmarking dataset frim sleep studies are used to give numerical evidence that this CMMN method leads to significant and consistent performance gains that are independent of the neural network architecture when adapting from a data collection device to another, data collection session to another, or even subject to another. These contributions form critical stepping stones in applying machine learning to health and medicine.

**Strengths:**

The paper introduces a new and efficient method to normalise biomedical signals, especially EEG. It experiments of an established benchmarking dataset frim sleep studies are used to give numerical evidence that this CMMN method leads to significant and consistent performance gains that are independent of the neural network architecture when adapting from a data collection device to another, data collection session to another, or even subject to another. These contributions form critical stepping stones in applying machine learning to health and medicine.

**Weaknesses:**

The paper has quite lacking referencing of related methods to de-noise, normalise, and otherwise pre-process EEG data. For example, I was expecting to see HAPPE from Harvard mentioned. See https://www.sciencedirect.com/science/article/abs/pii/S1388245723000275?via%3Dihub for a recent study with such pointers to not only HAPPE but also the open challenges in EEG processing from neuroscience perspectives.

I would the study be so specific and focesed on biomedical signal processing that its contributions to mechine learning are only fair. In addition to strengtening the articulation of methodological contributions to machine learning (currently this part is quote resticted, shallow, and repetative), the study calls for a more extensive discussion from machine learning perspective. Here, for example, the broader perspective of better abilities to adapt from data collection processes or subjects to others could be revisited, deepened, and connected with related studies, thereby releasing space in the beginning of the paper for making a more convincing case as a machine learning study.

**Questions:**

What are the best/good practice workflows in sleep studies or other biomedical studies with EEG data to normalise and then, eg, classify or regress using neural networks?

Why is this study well positioned to, eg, machine learning researchers? What can a reader not working on EEG gain from the study?

**Limitations:**

As a minor comment, it would be good to add a small ethical considerations addressing, e.g., in the discussion section. This is because data from human subjects is in use in this paper, and hence, even if not mandatory, addressing human research ethics is important.

---

> ### Author Rebuttal · Authors · 2023-08-08
>
> > The paper has quite lacking referencing of related methods to de-noise, normalise, and otherwise pre-process EEG data. For example, I was expecting to see HAPPE from Harvard mentioned. See https://www.sciencedirect.com/science/article/abs/pii/S1388245723000275?via%3Dihub for a recent study with such pointers to not only HAPPE but also the open challenges in EEG processing from neuroscience perspectives.
>
> There are indeed several preprocessing pipelines that have been promoted for EEG. Interestingly we would like to refer to this recent work from Arnaud Delorme, “EEG is better left alone”, and to the methods compared in this work. As the RELAX paper suggested, this work is not machine learning as it has no prediction in mind. It evaluates effect size with ANOVA to compare two experimental conditions. Again it is not about prediction on unseen subjects and domains. Nonetheless, our approach should not be seen as a replacement of those pipelines but as an additional adaptation step that can be used jointly. Actually, our work is more a novel way to filter data in the time domain, which offers significant prediction performance boosts when combined with a deep learning approach to sleep data. We will discuss and clarify this in the final version of the manuscript and cite HAPPE, RELAX, and other works evaluated by Delorme.
>
>
> > I would the study be so specific and focesed on biomedical signal processing that its contributions to mechine learning are only fair. In addition to strengtening the articulation of methodological contributions to machine learning (currently this part is quote resticted, shallow, and repetative), the study calls for a more extensive discussion from machine learning perspective. Here, for example, the broader perspective of better abilities to adapt from data collection processes or subjects to others could be revisited, deepened, and connected with related studies, thereby releasing space in the beginning of the paper for making a more convincing case as a machine learning study.
>
> This paper proposes a novel methodological approach (on Monge barycenter and subject-based mapping) and an in-depth evaluation of a challenging clinical task. We wanted to offer the necessary ML background while providing convincing experiments, thereby reaching a wider audience. We will happily add suggested references beyond the one already provided in the DA literature.
>
>
> > Questions:
> > What are the best/good practice workflows in sleep studies or other biomedical studies with EEG data to normalise and then, eg, classify or regress using neural networks?
>
> For sleep staging, the usual practice is to do very little preprocessing. Some papers use (high and low pass) filtering and scaling, some others use only scaling. The idea is that deep learning is doing all the normalization automatically, provided it is trained with enough data. For example, DeepSleepNet (Supratak et al. 2017) does not suggest any data normalization. Works like HAPPE or RELAX have a strong focus on artifact rejection like blinks which are not adapted for sleep (actually, blinks are signals for REM sleep). Here we followed standard practice in sleep deep learning works.
>
>
> > Why is this study well positioned to, eg, machine learning researchers? What can a reader not working on EEG gain from the study?
>
> Note that the paper was submitted to the neuroscience track because we know that it will be mostly of interest in this field. On the other hand, the method is quite generic for any problem of supervised learning on time series and can be applied for multi-source time series problems. Besides this work details a novel mathematical result and algorithmic contribution in addition to an experimental study. Hence, we strongly believe that NeurIPS is a good venue for presenting it.
>
>
> > As a minor comment, it would be good to add a small ethical considerations addressing, e.g., in the discussion section. This is because data from human subjects is in use in this paper, and hence, even if not mandatory, addressing human research ethics is important.
>
> Good point. All datasets used in our experiments are anonymized and public datasets that have already passed an ethics committee before recording. Still, this kind of application raises questions about the personalization and personal data that will be used, so we will indeed add a discussion in the final version.

---

> > ### Comment · Reviewer_Dgon · 2023-08-15
> > **Rebuttal response**
> >
> > Based on the rebuttal responses to address reviewers' concerns, I have revised my scoring.

---

### Official Review · Reviewer_crXL · 2023-07-06

**Soundness:** 2 fair
**Presentation:** 2 fair
**Contribution:** 2 fair
**Rating:** 4
**Confidence:** 4

**Summary:**

The paper presents a method for the spectral normalization of a training set of stationary time-series to their PSD barycenter under the Fréchet or Bures-Wasserstein divergence, which relies only on first and second moments. Spectral filtering is applied to normalize unseen time-series at test, which should improve performance of models trained on the normalized data. Results on sleep-stage classification using existing architectures and data show consistent improvement in performance indicating better generalization.

**Strengths:**

The paper is fairly clear and the approach is very straightforward.

The paper's results are of practical importance.

The paper presents a method that could be seen as a common spectral filtering. Implying that the relative spectral power in a subject/session is better than absolute power in cases of sleep stage classification. This is an interesting conclusion.

The experiments go into detail to show that results are consistent across model architectures at both the session and subject level.

**Weaknesses:**

My main concern is that the experimental results are only tested in sleep staging. It is intuitive that this normalization helps in tasks where the relative frequency content is important but that it may vary due to difference in setup. But it is not clear what other EEG analysis paradigms it would help with such as in the analysis of evoked potentials, steady-state evoked responses, motor imagery classification, etc. Likewise, it is not clear the spectral filtering would improve other non-EEG biosignals as mentioned in the title. EEGs are a special case as compared to ECG or other more structured biosignals. The claims are not backed by evidence.

The discussion could be improved by discussing the physical interpretation on differences of spectrum such as differences in impedance of the electrode setup between subjects or between sessions.

The importance of the barycenter as the target of the common normalization of the spectral filtering is not clear. A simple baseline would be to define a target PSD for the signal for each subject as a power law of the frequency across the frequency range of interest (~1 Hz to 30 Hz). This is not a barycenter but instead a fixed target that could also be implemented efficiently. Note that a power law with exponent of zero would be whitening the signal.

The paper could make it more clear that the equalization is done at the subject basis. It isn't stated directly and seems to allude to multiple choices.

Minor points:
Line 119 would be more precise by saying that the processes are zero mean.

The observation about the connection to Hellinger in line 140 is indirect. It is well known that for the Bures distance underlying the Fréchet distance that one can compute the distance directly in terms of the eigenvalues if the covariance matrices commute.


**Questions:**

1. Would it work for different EEG tasks? If not what are the limitations (spectral normalization may not be as important for evoked responses in an odd-ball paradigm for example)?

2. Would a power law target (including whitening) for the equalization perform as well in practice for EEG? I don't think there is any theoretically optimal reason that the barycenter should help downstream tasks, even though it minimizes the average divergence.

3. Line 53 and 54 It is not clear the particular import of the statement about source data availability  "which might not be available in practice due to privacy concerns or the memory limit of devices." Can this be further explained?


**Limitations:**

The paper's limitation are its application to only a specific task of EEG where relative frequency is known to be important.

---

> ### Author Rebuttal · Authors · 2023-08-08
>
> > My main concern is that the experimental results are only tested in sleep staging. It is intuitive that this normalization helps in tasks where the relative frequency content is important but that it may vary due to difference in setup. But it is not clear what other EEG analysis paradigms it would help with such as in the analysis of evoked potentials, steady-state evoked responses, motor imagery classification, etc. Likewise, it is not clear the spectral filtering would improve other non-EEG biosignals as mentioned in the title. EEGs are a special case as compared to ECG or other more structured biosignals. The claims are not backed by evidence.
>
> > Would it work for different EEG tasks? If not what are the limitations (spectral normalization may not be as important for evoked responses in an odd-ball paradigm for example)?
>
> The focus on Sleep stage data is discussed in the global response. Indeed CMMN is focusing on frequency content in signals which is well adapted to sleep data that use few electrodes. Indeed it is likely not adapted to other EEG setups like motor imagery or ERP using many electrodes covering the full scalp surface.
>
> Nevertheless we would like to emphasize that our results definitely show that CMMN improves on the state-of-the-art in some applications of EEG processing and we believe that it is already valuable for the community. Besides, our work integrates very well with existing deep learning methods with very small computational cost. In our experiments CMMN brings a systematic performance boost. The fact that our approach might be of limited use in other setups should not solely justify rejecting the paper. As discussed in the general reply we agree that the title could be changed to reflect the contribution better.
>
> > The discussion could be improved by discussing the physical interpretation on differences of spectrum such as differences in impedance of the electrode setup between subjects or between sessions.
>
> Thank you for this suggestion. Using subject-dependent filters as in CMMN allows Indeed to compensate for different impedances that can be modeled physically by a convolution. We will add a discussion to the paper.
>
> > The importance of the barycenter as the target of the common normalization of the spectral filtering is not clear. A simple baseline would be to define a target PSD for the signal for each subject as a power law of the frequency across the frequency range of interest (~1 Hz to 30 Hz). This is not a barycenter but instead a fixed target that could also be implemented efficiently. Note that a power law with exponent of zero would be whitening the signal.
>
> > Would a power law target (including whitening) for the equalization perform as well in practice for EEG? I don't think there is any theoretically optimal reason that the barycenter should help downstream tasks, even though it minimizes the average divergence.
>
> This is a very good point. We discuss in the global response new experiments that compare CMMN to whitening and  Power law target PSD. CMMN outperforms those target PSD in those experiments. Besides power law target and whitening lead to higher variance in across subjects results.
>
> Also, note that our proposed barycenter is non-parametric (power law parameters need to be selected), and can be computed efficiently on the source data with the new closed-form we proposed.
>
> > The paper could make it more clear that the equalization is done at the subject basis. It isn't stated directly and seems to allude to multiple choices.
>
> Indeed, our paper proposes a method for multi-source problems that performs a different Monge filtering for each source and target domain. The sources are indexed by k in the equations. Depending on the experiments, a domain can be a session or a subject. We will clarify that we consider subjects as sources in the experiment for the final version.
>
> > Minor points: Line 119 would be more precise by saying that the processes are zero mean.
>
> Agreed. We will do that. Thanks.
>
>
> > The observation about the connection to Hellinger in line 140 is indirect. It is well known that for the Bures distance underlying the Fréchet distance that one can compute the distance directly in terms of the eigenvalues if the covariance matrices commute.
>
> Not sure what you mean. It is a known relation related to Frechet distance in the space of covariance, but we found it interesting. Let us know how you would suggest we rephrase or remove this observation.
>
> > Line 53 and 54 It is not clear the particular import of the statement about source data availability "which might not be available in practice due to privacy concerns or the memory limit of devices." Can this be further explained?
>
> In medical fields, data is private and hard to share because of privacy concerns. Moreover, hospitals can have limited computational resources (memory, GPUs ...). Our method can use pre-computed PSD and does not require further training. A new hospital can simply apply a trained model without the need to have access to the source data (only to the barycenter) which is a problematic requirement for many alternative domain adaptation methods.

---

> > ### Comment · Reviewer_crXL · 2023-08-14
> > **response to rebuttal**
> >
> > First, I want to thank the authors for the rebuttal. I have read it and looked over the original submission.
> > **General applicability to EEG especially multichannel set-ups**
> > I think the method could or should be tested on other paradigms besides sleep staging. I'm not satisfied by the statement that "indeed it is likely not adapted to other EEG setups like motor imagery or ERP ". One could imagine that a single filtering is done by averaging pooling the spectra across channels. In this case the spatial information would be preserved for multichannel signals.  As I mentioned the hypothesis for convolutional Monge mapping is that it makes the representation invariant to what are likely non-informative differences in absolute spectral content.
> >
> > I'm not saying that limited scope is a reason for rejection. I'm just concerned about the scope of claims that are either too expansive like the original title, or too limited like the new statement that it is not applicable to multichannel recordings.
> >
> > **Relationship between new target spectrum and variance**
> > Thank you for running the suggested baselines. I do agree that the barycenter is a more natural normalization structure (as the actual data informs the filtering), but it is interesting that there are some cases the other normalizations perform better. My hypothesis is that regularization of the filtering (akin to Laplace smoothing for probability mass function estimation) would reduce variance.
> >
> > **Statement about Hellinger**
> > I agree that this is a special case. Nonetheless, upon rereading the original statement I still agree it is indirect and think the sentence can simply be restated to be much clearer. For example, the pronoun "it" is ambiguous, as is the construction of the sentence.
> >
> > Also, the PSD for a given signal is estimated by taking the $\ell_2$ average of short-time discrete Fourier transforms (STDFT), but the barycenter among multiple processes uses the Hellinger barycenter, which as mentioned is robust to outliers. What would the implications of forming a barycenter of the STDFT for each process?
> >
> > **New comments**
> > While reviewing the submission I found the statement "for a long enough signal, one can assume that the signal is periodic," is somewhat misleading. This is not the usual treatment of the discrete Fourier transform. Sampling in frequency is done for signals where the time-domain signal to be time-limited, possibly by a windowing function. In the case of auto-correlation function/sequence the time limiting happens naturally for aperiodic signals when the correlation structure goes to zero for large lags. Thus, the auto-correlation matrix, which is by design Toeplitz is extended to be circulant through the imposition of a periodic boundary condition necessary for the discrete Fourier transform.
> >
> > The use of windowing functions in conjunction with Welch's method should be clarified as the assumption of a rectangular window is not always desirable. Also typically there is a division by the sum of the window squared https://www.osti.gov/servlets/purl/5688766. If the same windowing function is used this constant cancels out.

---

> > > ### Author Response · Authors · 2023-08-15
> > > **Thank you very much for your response**
> > >
> > > First we want to thank the reviewer for his time and for this interesting discussion that will make our paper better.
> > >
> > > **General applicability to EEG especially multichannel set-up**. Our response was probably a bit abrupt. We did not mean that the method was useless in a multi-sensor configuration but that we believe that it will bring less since spatial information usually helps for the end task and we do not address that with independent filters. We agree that it is a very interesting question that deserves some time and space, which we do not have in the current paper. We plan on investigating this in future work along Monge spatio-temporal filtering and investigate the performance of the current CMMN method with such extensions. We will discuss this in the final version but we believe that it is clearly out of the scope of the current paper that already contains many experiments to illustrate the behavior of the method and prove its interest on sleep stage classification.
> > >
> > >
> > > **Relationship between new target spectrum and variance**. Regularization of the filtering is indeed an interesting idea and  a classic approach on EEG (also spatial filtering) but might be hard to integrate with or fast FFT solvers. We believe that the filter size is an indirect regularization of the filters that are smooth by construction in the frequency domains with small sizes (resolution). Additional regularization might be done by adding to the barycenter a virtual weighted  domain with a “smooth” PSD. But this also adds a parameter that potentially has a large impact on performance and that is hard to select  when no labels are available and that probably depends on the target domains.
> > >
> > >
> > > **Statement about Hellinger**. We will reformulate the statement as suggested. The reason why we estimate PSD for each domains with L2 is because we wanted to use the classical Welch estimator that is widely available but we agree that computing the domain PSD with the Monge/Hellinger barycenter instead is interesting and we will add an experiment with this to the camera ready version of the paper. Interestingly this would mean that the final barycenter would be the solution of a hierarchical OT problem (OT barycenter of OT barycenters).
> > >
> > >
> > > **New comments**. We will reformulate the statement about short time FFT as suggested by discussing the limited support correlation (that goes to 0 for large lags) of aperiodic signal and the circulant Toeplitz approximation. Thank you for the suggestion.
> > > We will also clarify that we use a rectangular window in the experiments. In our experiments we have tested other windows (Hann) and found very small changes in performances of CMMN, we will add those results in supplementary.
> > >
> > >
> > > As a small conclusion we believe that our submission can already bring a lot to the community and we demonstrated with the new results that the method is of definite interest for the application at hand. The experiments on new data suggested by the reviewer are very interesting and will be investigated but would not fit in a conference paper. We believe it is better for the community to do one application in depth (sleep stage classification) than several shallow applications and proof of concepts. One thing that is clear for us from this very interesting discussion with reviewer crXL is that this seminal work on convolutional Monge mapping for subject and session specific adaptation is stimulating, has a lot of potential impact and opens the door to novel research ideas. This is in our opinion what makes a good NeurIPS paper.

---

### Official Review · Reviewer_VNrC · 2023-07-27

**Soundness:** 3 good
**Presentation:** 3 good
**Contribution:** 3 good
**Rating:** 6
**Confidence:** 4

**Summary:**

The paper proposes a normalization scheme for biological signals such that the model generalizes well to unseen domains which slightly deviate from the training data.
The approach is based on learning a normalization scheme from the training set in a heuristic manner and applying it at inference.

**Strengths:**

- Very intuitive and straight-forward method to handle distribution shifts.
- The idea behind the paper is of great importance to the community as there is a trend towards building more robust models rather than just pushing state-of-the-art. This plays a noticeable role in the reviewer's score justification and the reviewer would be eager to adjust their score after the rebuttal properly addresses the weakness/questions below.

**Weaknesses:**

- The contribution is relatively mild compared to [27].
- The proposed method, just like any other normalization scheme, has a failure threshold not explored by the authors.

Minor:
- Figure 1 is very difficult to read. The reader sees this figure without having any idea about what Eq. (5) and (6) are. Typically, the figure that appears at the 2nd page is a either a summary of the pipeline in a easy-to-follow format or a summary of the main result of the paper. Currently, Figure 1 serves no purpose but misleading and confusing the reader.
- Line 133: It's either a proof or sketch of the proof, can't be both.



**Questions:**

- It's not clear how much of a divergence is tolerated by Monge mapping. More precisely, what happens if the test signal deviates more than a threshold from the source domains? Would forcing such normalization be harmful in that case?

---

> ### Author Rebuttal · Authors · 2023-08-08
>
> > The contribution is relatively mild compared to [27].
> > The proposed method, just like any other normalization scheme, has a failure threshold not explored by the authors.
>
> We respectfully disagree with the reviewer w.r.t. the contribution compared to [27]. Reference [27] is mostly theoretical with statistical results on the convergence of the Monge mapping and domain adaptation bounds. The experiments are limited to toy examples and a particular case of MNIST. In our work, we indeed use the linear convolutional mapping from [27] but also propose a novel closed-form formula for the barycenter and extend the mapping/domain adaptation to the multi-source setting in a non-obvious way. Besides, our experiments are done in-depth on several large datasets in a challenging biomedical problem.
>
> We discuss failure and limits below and report new studies about performance VS distance and kurtosis in the general reply.
>
> > Minor:
> >
> > Figure 1 is very difficult to read. The reader sees this figure without having any idea about what Eq. (5) and (6) are. Typically, the figure that appears at the 2nd page is a either a summary of the pipeline in a easy-to-follow format or a summary of the main result of the paper. Currently, Figure 1 serves no purpose but misleading and confusing the reader.
>
> The idea of Fig. 1 was to illustrate the mapping and adaptation of the signal samples for each subject in the PSD space. We will amend the figure with the whole pipeline for the final version of the paper, which should be less confusing.
>
>
> > Line 133: It's either a proof or sketch of the proof, can't be both.
>
> Indeed we will correct that. Thanks for noticing.
>
>
> > Questions:
> > It's not clear how much of a divergence is tolerated by Monge mapping. More precisely, what happens if the test signal deviates more than a threshold from the source domains? Would forcing such normalization be harmful in that case?
>
> That is an interesting question. To discuss it, we propose new plots showing the evolution of the BACC vs Bures-Wasserstein distance. This is discussed in more detail in the global response.

---

> > ### Comment · Reviewer_VNrC · 2023-08-18
> >
> > Thanks for the reply to my comments and making significant efforts in the rebuttal. Although the first point regarding the contribution is still a place of debate, new changes have added more value to the paper and that's why I've raised my score.

---

### Author Rebuttal · Authors · 2023-08-08

We want to thank all reviewers for their interesting questions and suggestions. We provide in the attached PDF new results (figures and tables) that we refer to in our detailed answers below. First, we make a few general comments that we believe are of interest to all reviewers.

### Focus on Sleep Staging data

We believe that the clinically-relevant problem of sleep staging is an ideal ML task for studying the question of subject-based domain adaptation on biosignals. Biosignals suffer from large intersubject variability that is reflected in sleep EEG data. Nowadays, sleep data are easy to access and come with abundant labels enabling convincing experimental evaluations. Again, this makes sleep staging from EEG a perfect setup to compare DA methods on biosignals.

While there exist other applications of ML on EEG (BCI MI and ERP), they usually contain few subjects which leads to large error-bars when evaluating algorithms across subjects (SoTA Riemannian approaches on the reference BCI4a dataset gives a 10% error bars with 9 subjects). That greatly diminishes the interest and the statistical significance of results of multi-source DA. Sleep data with thousands of labels and hundreds of subjects are a better testbed for methods like CMMN.

In addition, CMMN leverages spectral information and efficient FFT-based computations. Sleep EEG data do have a predictive spectral content.  In other biomedical applications on signals, the predictive power comes from spatial components which implies that many sensors are available at different locations. In sleep EEG we have here only 2 channels to keep the experimental setup simple in a clinical context. Future work will consist in extending CMMN to make use of spatial diversity in sensing.

Yet, we agree that the current title can be amended and we are OK to ask the PC/AC to slightly update the title to replace “biosignals” with “sleep EEG signals” to better reflect the experimental results.

### Effect of the distance between domains

Figure 1 shows a new visualization of the BACC versus the Bures-Wasserstein distance. As we can see, with no normalization, the BACC always decreases with the distance. CMMN reduces this effect drastically. In fact, for each adaptation between domains, the slope is reduced with CMMN and even canceled for SHHS to MASS.

This study suggests that CMMN is less dependent on the distance of the domain to the barycenter and that the Monge mapping can compensate for even large shifts.

### Study of non-gaussianity

Figure 2 plots the BACC versus the kurtosis for no normalization and CMMN. These plots do not allow to detect any dependence between kurtosis (as a proxy for non-gaussianity) and performance either for CMMN or no normalization. This interesting result suggests that the gaussian/linear models in the methods do not limit the performances, maybe because higher order information that can be used by the CNN for classification is fully preserved.

### Interest of the OT barycenter

To demonstrate the importance of choosing the right target PSD when estimating CMMN mapping, we tested Monge mapping for different target PSD (power law with $a = 0.659$ and whitening). The results are shown in table 1.

As we can see, the suggested adaptations are interesting because some adaptations with whitening or power-law give good results (i.e., MASS->SHHS, Physionet->MASS, and SHHS->SHHS). However, on average, choosing the barycenter outperforms other target PSD (2% better in BACC and 1.5% better in $\Delta$BACC). Furthermore, for one of the hardest adaptations (SHHS -> MASS/Phys.) taking another target PSD leads to a very high variance in the scores (~ 13% of std). We will update the paper and thank the reviewer for suggesting this interesting new experimental result to strengthen our claims and paper.

---

### Decision · Program_Chairs · 2023-09-21

**Decision:**

Accept (poster)

**Comment:**

This paper introduce a novel OT-based method for domain (test-time) adaptation, targeting learning on biosignals. The work is motivated and presented well, thoroughly demonstrates its benefit empirically and reviewers consider the work of of practical importance to the community while highlighting the relative simplicity and efficiency of the approach.